# A Transformer-Based Neural Network with Improved Pyramid Pooling Module for Change Detection in Ecological Redline Monitoring

Yunjia Zou [1,2], Ting Shen [1], Zhengchao Chen [1,*], Pan Chen [2,3], Xuan Yang [3] and Luyang Zan [1]

1   Airborne Remote Sensing Center, Aerospace Information Research Institute, Chinese Academy of Sciences, Beijing 100094, China
2   University of Chinese Academy of Sciences, Beijing 100049, China
3   Key Laboratory of Digital Earth Science, Aerospace Information Research Institute, Chinese Academy of Sciences, Beijing 100094, China
*   Correspondence: chenzc@aircas.ac.cn

**Abstract:** The ecological redline defines areas where industrialization and urbanization development should be prohibited. Its purpose is to establish the most stringent environmental protection system to meet the urgent needs of ecological function guarantee and environmental safety. Nowadays, deep learning methods have been widely used in change detection tasks based on remote sensing images, which can just be applied to the monitoring of the ecological redline. Considering the convolution-based neural networks' lack of utilization of global information, we choose a transformer to devise a Siamese network for change detection. We also use a transformer to design a pyramid pooling module to help the network maintain more features. Moreover, we construct a self-supervised network based on a contrastive method to obtain a pre-trained model, especially for remote sensing images, aiming to achieve better results. As for study areas and data sources, we chose Hebei Province, where the environmental problem is quite nervous, and used its GF-1 satellite images to do our research. Through ablation experiments and contrast experiments, our method is proven to have significant advantages in terms of accuracy and efficiency. We also predict large-scale areas and calculate the intersection recall rate, which confirms that our method has practical values.

**Keywords:** deep learning; change detection; transformer; pyramid pooling; self-supervised study; ecological redline; GF-1 satellite

## 1. Introduction

Nowadays, with the rapid development of China's urbanization and industrialization, the environmental problem has become increasingly severe. Although the government has tried hard to increase the monitoring and protection of the ecological environment, the deterioration of the overall environment has never been curbed. To deal with this ecological crisis, the ecological redline (Eco-redline, ECR) policy is proposed, which aims to maintain vital ecosystem services needed for sustainable social development through coordinated nationwide planning. The policy defines key ecological protection areas where five necessary national ecosystem services [1] should be maintained: flood disaster mitigation, sandstorm disaster prevention, water resources protection, soil resources conservation, and biodiversity conservation. To sum up, the Eco-redline represents an attempt to establish strict criteria for assessing ecosystem services in land-use planning, and is defined as "the minimum ecological area for the protection of the safety and functioning of the ecological environment and the maintenance of a country's biodiversity." [2].

To properly regulate the Eco-redline areas, large amounts of data and advanced technology are needed. In recent years, remote sensing datasets acquired by multiplex spaceborne and airborne sensors with rich temporal, spatial, spectral, and radiometric

resolution characteristics have largely increased [3]. Due to the rapid development of sensors and the advancement of image processing technology, the increasing trend of remote sensing big data will surely continue [4].

Based on the ever-increasing data, change detection using remote sensing images has surpassed the means of manual field exploration and has become one of the most common research techniques. This technology can function in pixel-level and grid-level research, which leads to wide use in urban monitoring, environmental monitoring, and post-disaster damage assessment [5,6]. Since it can quickly and accurately explore changes in natural landscapes, it can provide real-time feedback on changes in mountainous areas and woodlands, which perfectly fits the research needs of detecting human-induced changes in Eco-redline areas.

Therefore, many remote sensing change detection methods have been proposed, some of which have been tested and validated in many studies [7–10]. For example, traditional methods such as principal component analysis (PCA) [11], independent component analysis (ICA) [12], and multivariate alteration detection (MAD) [13] have been successfully applied to many change detection studies. However, given the interference of many internal factors on change detection, these methods still cannot be reliably applied to practical situations. Moreover, since the purpose of change detection is to quantitatively analyze and determine the surface changes in different periods through remote sensing data, the following problems cannot be avoided: firstly, there are too many land cover categories, and the similarity between different sorts and the diversity between the same sorts will disturb the research. Secondly, systematic errors such as the interference of the imaging environment and the sensing system will lead to huge differences in the remote sensing data obtained from different sensors in the same area. Finally, there are seasonal changes in the land cover itself, such as various grasslands and woodlands changing with the seasons, and these irrelevant changes will no doubt hinder the extraction of research targets. Therefore, it is quite difficult to obtain highly accurate change results to meet the requirements of Eco-redline monitoring.

Fortunately, the recent remarkable achievements of neural networks in the field of computer vision have contributed to the rise and development of deep learning-based change detection methods [14–20]. Such methods have strong feature expression capabilities and can extract more key information from images. Depending on whether the dataset has labels, there are two types of deep learning methods, unsupervised and supervised. The purpose of the change detection method based on the unsupervised neural network is to learn a new feature space through the neural network so as to shorten the distance between different time-phase feature spaces. Although unsupervised learning improves the discriminative ability of new feature spaces to some extent by introducing pseudo-labels [20–22], the problem of detecting changes that are not related to the problem of interest still arises in the real application. Compared with unsupervised methods, the rich label information possessed by supervised neural network-based change detection methods can better distinguish the specific change types of interest in the whole scene but will need much more effort to finish the labeling job. Therefore, sometimes these two approaches are combined to achieve better results with less workload.

The following question is about the information fusion method used when constructing a change detection network. Generally, there are three levels of information fusion, namely data-level fusion, feature-level fusion, and decision-level fusion [23]. Among them, data-level fusion is the easiest way. It straightly concatenates the pre- and post-phases as the input of the network. This is the strategy adopted when applying many typical semantic segmentation networks to the field of change detection. However, sometimes this method may receive unexpectedly low-precision results [24]. Moreover, since two inputs only produce one output in change detection, decision-level fusion can not be possible [25]. Therefore, to achieve high-performance networks, more and more researchers have begun to use the idea of feature-level fusion, thus, giving birth to the Siamese network. The Siamese network inputs data of different phases into different network branches to

complete feature extraction. These high-level features will then be combined to achieve feature-level information fusion, and finally, the combined information will be used to identify the changes. The Siamese neural network proposed in this theory is usually regarded as the basic network structure for acquiring high-level features [26,27].

Then, the sampling method will also need to be considered. The traditional convolution is constrained by the limited receptive field and lacks the utilization of local information. To further optimize the network performance, many researchers have introduced Visual Transformer (ViT). The Multi Self-Attention (MSA) module of traditional ViT can well utilize global information based on ensuring parallel computing [28]: for example, Wang et al. combined the convolutional neural network with the CBAM attention mechanism to improve the feature learning performance of radar image change detection [29]. Peng et al. optimized and improved the UNet++ network by replacing the original upsampling unit using the attention mechanism [30]. This approach enhances spatial and channel attention guidance and achieves better results than the original network. Chen et al. propose an attention information module AIFM and combine it with the Siamese ResNet [25]. As a bridge of feature fusion, this module improves the performance of the network for feature extraction of changes in remote sensing images.

Since the self-attention mechanism is quite successful, how to effectively apply and further improve it has become a popular direction. Recently, the Hierarchicle Vision Transformer using Shifted Windows (Swin Transformer, Swin-T) was proposed and rapidly achieved remarkable results on many experimental tasks [31]. It uses the window-based multi-self-attention (W-MSA) module to replace the traditional ViT MSA module, which greatly reduces the amount of computation. To ensure the utilization of information, Swin-T additionally designs and adds a shifted window-based multi-self-attention (SW-MSA) module. The combination of W-MSA and SW-MSA can guarantee the interaction of global information. In addition, Swin-T introduces a relative position offset to increase the overall accuracy of the network further. Based on the idea of Swin-T, Swin-UNet was proposed in 2021 and was famous for its high efficiency as well as high performance [32]. It constructs a symmetric encoder-decoder structure with skip-connections based on Swin-T.

Given the excellent performance of Swin-UNet, our experiment considers modifying and applying it as the backbone to devise change detection networks. Actually, we use the feature-level fusion method and build a Siamese Swin-UNet for change detection called SWUNet-CD. However, to achieve the goal of obtaining higher-precision change results, the following problems will still need to be solved: firstly, the traditional pooling layers (such as the max pooling layer and average pooling layer) can inevitably lose some key features when applied to more complex remote sensing images [33–35]. Secondly, public pre-trained models are trained on the ImageNet dataset, and these images can be quite dinstinct from remote sensing images due to geographic information and shooting angles. Thirdly, to better deal with the problem of large-scale unbalanced distribution, especially in our research regions, we need to pay more attention to the research area and the aimed objects of our experiment. To solve the problems listed above, we make two major improvements: (1) we combine the W-MSA/SW-MSA mechanism with the idea of multi-scale fusion, and design a swin-based pyramid pooling module (SPPM) with a more complex structure. The purpose is to improve the feature expression ability of the network and retain more key features when applied to complex remote-sensing images. (2) We construct a self-supervised network using the idea of the contrastive method. The purpose of this network is to provide a more suitable pre-trained model for our downstream tasks. This network is trained on a large amount of unlabeled data in the study area, where pseudo-labels are produced by employing data augmentation. Its purpose is to reduce the influence caused by the differences between the datasets where negative samples are much more when compared to positive samples, and keep closer to our research area and aimed objects.

In summary, we build SWUNet-CD, a Siamese change detection network based on Swin-UNet. To better apply it in the remote sensing field, we design SPPM, a special

pooling module to replace the traditional pooling layer, and construct a self-supervised network that is used to obtain specialized pre-trained models. The main contributions of this paper are as follows:

- A Siamese self-supervised learning network is constructed using the idea of contrast. The original image and the data-augmented image are used as two inputs to the network, and the classification of each data augmentation is the output. By training with a large amount of unlabeled remote sensing data from the study area, a set of pre-trained models for loading into subsequent supervised networks is finally obtained. We have verified through experiments that this method helps the proposed network better adapted to remote sensing change detection;
- Based on the idea of Swin-T and multi-scale fusion, a self-attention mechanism-based pyramid pooling module SPPM is constructed and applied to the Siamese Swin-UNet network adopted in this experiment. The self-attention module can better utilize global information to obtain more complete features, while the multi-scale fusion method can maximize the use of the information of each pixel and reduce the loss of key features;
- We apply the Siamese Swin-UNet network using a special pre-trained model and improved pooling module SPPM to the problem of remote sensing change detection for ecological redline monitoring and verify the performance of this change detection method through experiments.

The following contents are organized as follows. Section 2 clarifies the study area and the preparation of the dataset in detail. Section 3 provides a quite detailed theoretical introduction to the change detection method proposed. Section 4 contains ablation experiments, contrast experiments, and the calculation of intersection recall rate for large-scale prediction results. These are all used to justify the validity and the superiority in terms of performance. Section 5 is a refined conclusion of our research work.

## 2. Dataset Details

### 2.1. Study Area and Data Source

To achieve the goal of our experimental task, we use two sets of GF-1 remote sensing data in different phases (one group from 2018, the other group from 2022) in Hebei Province and have labeled some of the data. The parameters of the GF-1 satellite are shown in Table 1, while the panorama and the labeled data coverage of remote sensing images in Hebei Province are shown in Figure 1a.

**Table 1.** GF-1 satellite parameters.

|  | Panchromatic Camera | Multispectral Camera |
|---|---|---|
| Spectral range | 0.45–0.90 μm | 0.45–0.52 μm<br>0.52–0.59 μm<br>0.63–0.69 μm<br>0.77–0.89 μm |
| Spatial resolution | 2 m | 8 m/16 m |
| Coverage width | 60 km | 60 km/800 km |
| Revisit period | 4 days | 4 days |

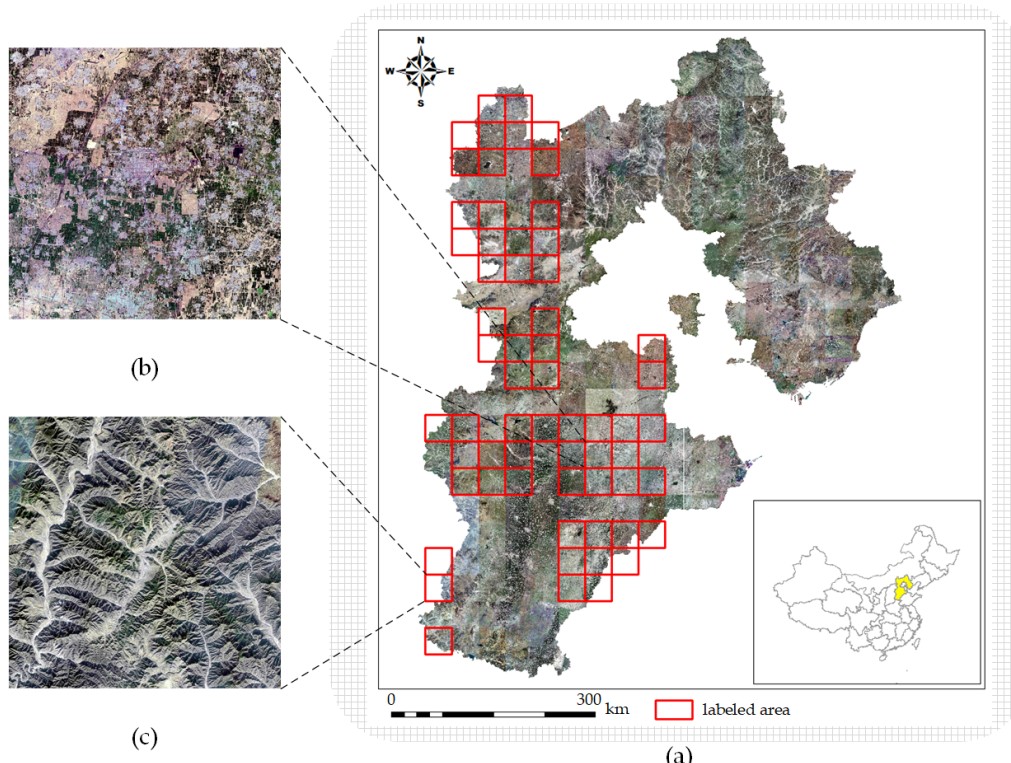

**Figure 1.** (**a**) Hebei Province and labeled area. (**b**) A remote sensing image from the city area. (**c**) A remote sensing image from Eco-redline area.

### 2.2. Labeling Method

The overall labeling work takes buildings, roads, construction sites, and mining areas as the main targets. The reasons are as follows: on the one hand, the urban areas consist of large numbers of buildings and roads whose changes can be easily identified. These changes can serve as typical samples to train the network preliminarily. On the other hand, since the damages to the Eco-redline areas, are mainly caused by illegal constructions of buildings and mining areas, the labeling work of these changes are more than vital because it straightly concentrates on the network's ability to monitor the Eco-redline areas. Changes in other ground objects, such as changes in cultivated land, are not labeled because their changes have nothing to do with our research goals. Therefore, we choose 61 groups of GF-1 satellite data in different phases to label, which are not geographically contiguous but suitable for our research. They have a spatial resolution of 2 m and a size of 16,384 pixels × 16,384 pixels. The detailed information of the labels, as mentioned above, only focuses on the changes that our experiment concerns, such as buildings, construction sites, and mining areas. To sum up, the overall dataset is distributed in two areas: one is the urban area (take Figure 1b as an example), where construction sites and newly added/removed roads and buildings are labeled. The other one is the Eco-redline area (take Figure 1c as an example), where the newly added/removed buildings and mining areas are labeled. These samples are combined to train the network, aiming to finally realize the target of intelligent information extraction of changes in Eco-redline areas.

### 2.3. Dataset Prepocessing

To facilitate the experiment, the samples need to be preprocessed to meet the needs of our network. In this task, we crop the four-band raw remote sensing images of 16,384 pixels × 16,384 pixels, TIFF format into three-band JPG images of 1024 pixels × 1024 pixels (we remove the near-infrared band in this process). The bands represent the three color channels of RGB, and the value range is (0, 255). At the same time, the corresponding manually labeled shapefiles are converted and cropped into 1024 × 1024 size, single-band

grayscale images. This single band represents the label value, including only two values of 0 and 1. After that, we make a detailed check and retain 3660 groups of samples with apparent changes, and remove the other 11,956 groups of samples without any variation. This work is to ensure the samples' equilibrium by reducing the proportion of backgrounds. The pre-phase, post-phase, and labels are named as A, B, and label, respectively. A and B are both three-band (RGB color channels) tensors when input into the network, and label is a single-band (label value) tensor when input into the network. We allocate these data into three datasets according to the ratio of 8:1:1, which are used for training, validation, and test, respectively.

## 3. Proposed Methods

In this work, we propose a deep learning network called SWUNet-CD for the change detection of the Eco-redline in Hebei Province using the GF-1 satellite images. Moreover, we devise a special pooling module and a dual-stream self-supervised network to improve its performance further. The detailed information is listed below.

### 3.1. SWUNet-CD

We convert the Swin-UNet into Siamese form for change detection, and apply the designed SPPM (mentioned in Section 3.2) to it. The main network structure of SWUNet-CD is shown in Figure 2a,b is a statistical summary of the names of different modules in it. The backbone of the network has four downsampling layers; each contains two consecutive Swin-T blocks, as shown in Figure 2c. Each Swin-T block consists of a normalization layer (LayerNorm), a MSA, a residual connection, and a Multilayer Perceptron (MLP). Among them, the Swin-T block in the front part uses W-MSA, and the latter part uses SW-MSA. The two consecutive Swin-T blocks can be expressed by the following formula:

$$\hat{x}^{2n} = W\_MSA(LayerNorm(x^{2n-1})) + x^{2n-1}, \tag{1}$$

$$x^{2n} = MLP(LayerNorm(\hat{x}^{2n})) + \hat{x}^{2n}, \tag{2}$$

$$\hat{x}^{2n+1} = W\_MSA(LayerNorm(x^{2n})) + x^{2n}, \tag{3}$$

$$x^{2n+1} = MLP(LayerNorm(\hat{x}^{2n+1})) + \hat{x}^{2n+1}, \tag{4}$$

$$Attention(Q, K, V) = SoftMax(\frac{QK^T}{\sqrt{Dimension}} + Bias)V, \tag{5}$$

among them, n represents the *n*th downsampling process, which includes the $(2n-1)$th Swin-T block and the 2*n*th Swin-T block. $\hat{x}^{2n-1}$ represents the output of the $(2n-1)$th MSA in the Swin-T Block, and $x^{2n-1}$ represents the output of the $(2n-1)$th MLP in the Swin-T Block. Self-attention is calculated as shown in Formula (5), where Q, K, and V denotes the query, key, and value matrixes, respectively. Dimension represents the dimension of the query or key, and the values in Bias are obtained from the bias matrix.

The functions of each module in the network are as follows: The function of the Patch Partition layer is to divide the input tensor into smaller pieces and then concatenate them in channel dimensions to shorten the tensor's length and width. The role of the Linear Embedding layer is to transform the channel size of the feature map obtained in the previous step to the size required by the network. The function of the Patch Merging module is to reduce the length and width of the feature map by half and double its channel size to form multi-level features. The Patch Merging module and the two consecutive Swin-T blocks jointly constitute a downsampling layer. The role of SPPM is to trim the downsampling results to remove the influence of overfitting while preserving the features as much as possible. The function of the Patch Expanding module is to double the length and width

of the feature map and reduce the channel size by half to restore the feature map hierarchically. The Patch Expanding module and the two consecutive Swin-T blocks together form an upsampling layer, where the feature information is fused by the corresponding downsampling layer through a skip connection. The role of the Linear Projection layer is to convert the feature map to a single channel for pixel-level change detection.

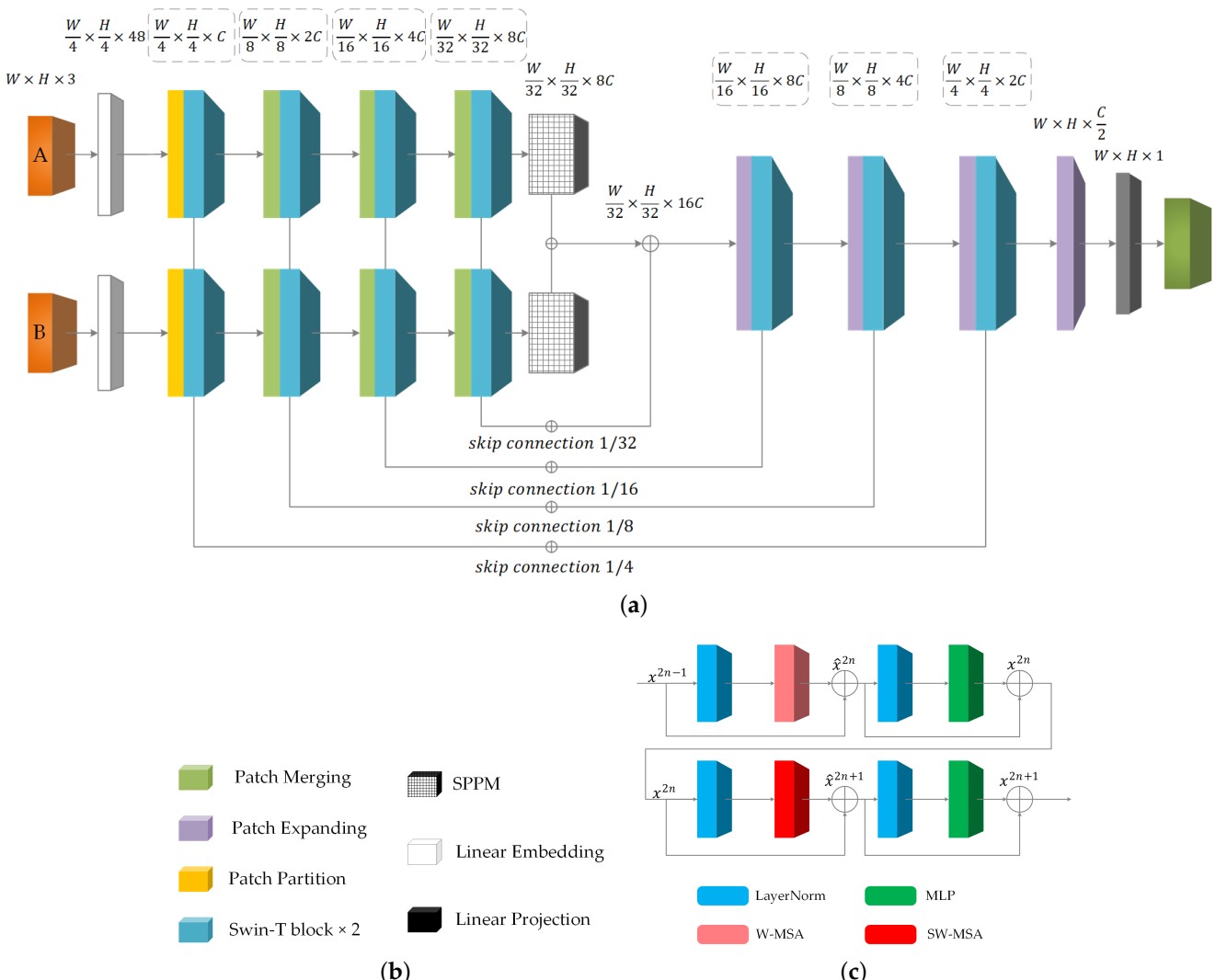

**Figure 2.** (**a**) The overall structure of SWUNet-CD; (**b**) the name of each layer; (**c**) the schematic diagram of two consecutive Swin-T blocks.

The overall process of our network is as follows: we input the pre- and post-phase images into the network. During the process of forward propagation, four downsampling operations and an SPPM-based pooling operation are performed on them, respectively. As a result, we obtain the features with downsampling four times, eight times, sixteen times, and thirty-two times. We then fuse the features of the same stage and use them as the input of the upsampling part. It is worth mentioning that the SWUNet-CD adopts the information fusion method of feature-level fusion instead of simple data-level fusion. Although this does increase a certain amount of parameter calculations, it can use deeper features to better train the network for the shallow features of remote sensing images that may usually be hard to detect and identify.

### 3.2. Swin-Based Pyramid Pooling Module

The idea of multi-scale fusion is innovative and has been proven to work well on the improvement work of the pooling layer, such as the Pyramid Pooling module (PPM) and the Atrous Spatial Pyramid Pooling (ASPP) [36,37]. Based on the multi-scale fusion idea and the excellent performance of W-MSA, we devise a Swin-based pyramid pooling module (SPPM), as shown in Figure 3. The pyramid consists of four feature extraction layers and one Linear Projection layer. Each feature extraction layer consists of a Patch Partition layer, a Linear Embedding layer, and two consecutive Swin-T blocks. The overall process of the entire SPPM is as follows: The Patch Partition layer downsamples the input feature map by two times, four times, eight times, and sixteen times, respectively. After that, these feature maps will go through the Linear Embedding layer in which their channel size will be transformed to C, $2 \times C$, $4 \times C$, and $8 \times C$ (C is defined by the hyperparameter as a channel size that the module can accept), respectively. These preprocessed feature maps are then input into the two consecutive Swin-T blocks, where the self-attention calculation is conducted on a window with a length and width of $7 \times 7$, defined by hyperparameters for each feature map. The windows' working mechanism is as follows: firstly, a self-attention calculation for a standard window is done. After that, another self-attention calculation is done for different partitions after the window is moved. The purpose is to achieve information exchange between different locations to maintain the utilization of global information after partitioning. Secondly, a mask calculation is used so that the self-attention between the moved part and the original part will not be taken into account. After passing through the Swin-T block, the output feature map of each layer structure will restore the length and width to the input size by reshaping operation. Later, they will be concatenated in the channel dimension. At the end of the module, the concatenated feature map goes through a Linear Projection layer to restore the channel size to the input size. In this way, SPPM can retain more deep features extracted by the network during the downsampling process, thus, avoiding losing some important information.

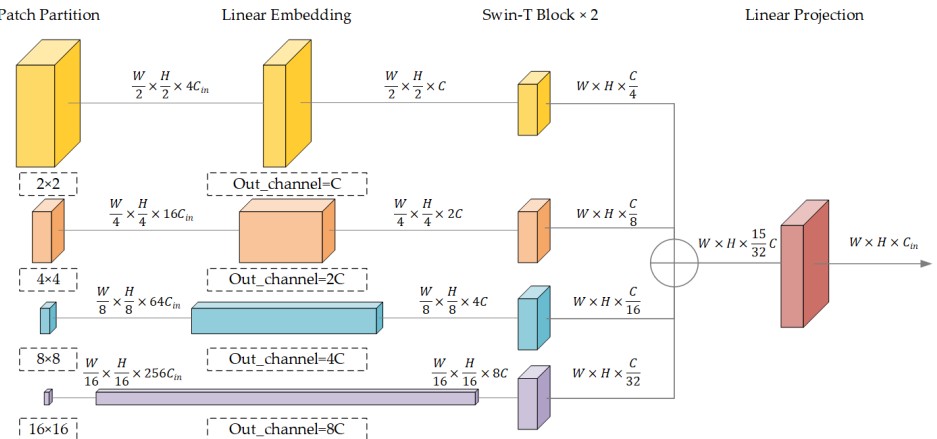

**Figure 3.** The schematic diagram of Swin-based Pyramid Pooling Module (SPPM).

In the SWUNet-CD built for our change detection experiment, we use this SPPM in its bottleneck layer. The purpose is to use the excellent performance of the self-attention mechanism in the field of feature extraction to make greater use of local and global information to retain multi-level information. As a matter of fact, we expect that this structure can further improve the function of the network compared with typical multi-scale fusion modules (such as PPM, ASPP, etc.) and obtain experimental results with higher accuracy.

### 3.3. Pre-Trained Model Obtained by Self-Supervised Network

Although our team has done some labeling work, it is still very scarce when compared to the overall remote sensing data. Therefore, we consider designing a self-supervised learning method. This method aims to construct several reasonable artificial labels for the

unlabeled remote sensing images and guide the network to learn more feature expressions that are helpful for the overall experimental task [38–42].

There exist two kinds of methods when constructing the self-supervised network: the generative method as well as the contrastive method. The former has many redundant parameters and is hard to optimize, so we adopt the latter. The contrastive method does not require pixel-by-pixel reconstruction of the input data. It only needs that the model can distinguish between different inputs in the feature space. Therefore, the network constructed in this way does not need a decoder. In fact, only a fully connected layer is acquired to judge the features obtained by the encoder. To better fit the downstream task, we construct a weight-sharing Siamese network, using Swin-T as its encoder, as shown in Figure 4. We construct artificial labels based on data augmentation. This idea is mainly derived from the phenomenon that the ground objects (forest, mining areas, illegal buildings, etc.), which are the objects of the study, are more distinguishable in color and texture. The original data and the augmented data are used as the dual-stream input of the Siamese self-supervised network. After acquiring both feature maps and comparing the differences, the predictive ability of the network is trained. Ultimately, the network can improve its ability to express features and produce a pre-trained model for the latter task.

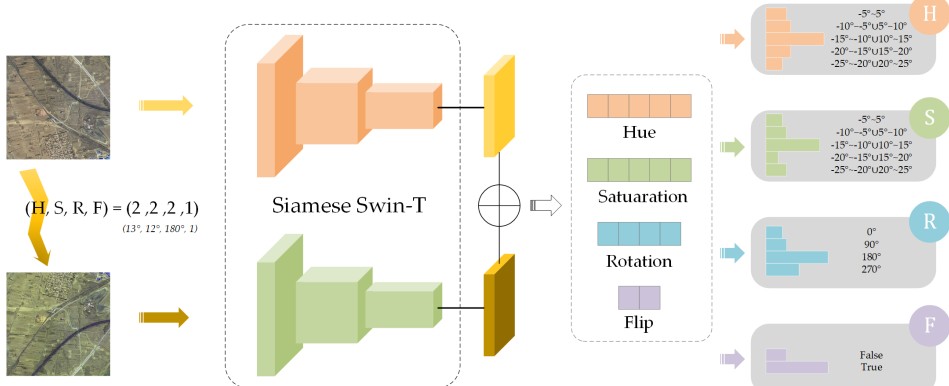

**Figure 4.** The schematic diagram of the duel-stream self-supervised network.

The contruction method is as follows: we use four data enhancement parameters as the judgment criteria, namely Hue, Saturation, Rotation, and Flip. Hue represents the relative lightness and darkness of the image, and its value ranges from 0° to 360°. Five types of transformation are used for Hue, and the ranges are $(-5°, 5°), (-10°, -5°) \cup (5°, 10°), (-15°, -10°) \cup (10°, 15°), (-20°, -15°) \cup (15°, 20°), (-25°, -20°) \cup (20°, 25°)$, respectively. These five types of ranges are labeled as 1–5. Saturation represents how close a color is to a spectral color when viewed as the spectral color mixed with white, and its value ranges from 0° to 100°. Five types of transformations are used for Saturation, and the ranges are $(-5°, 5°), (-10°, -5°) \cup (5°, 10°), (-15°, -10°) \cup (10°, 15°), (-20°, -15°) \cup (15°, 20°), (-25°, -20°) \cup (20°, 25°)$, respectively. These five types of ranges are labeled as 1–5. Rotation represents the rotation of the image by a certain angle. Four different rotation angles are used, namely 0°, 90°, 180°, and 270°. These four types of ranges are labeled as 1–4. Flip stands for flipping the image left and right. There are only two cases, including not flipping and flipping. These two types of ranges are labeled as 1 and 2.

The training process of the network is as follows: the original image and the data-augmented image are input synchronously into the network. After the down-sampling operation of Swin-T, two feature maps of the same size can be obtained. They are concatenated in the channel dimension to obtain the change information, which represents the data augmentation, and then this mixed feature map passes through a $1 \times 1$ size adaptive average pooling layer to extract the global information. Finally, four different classification results are acquired through four different fully connected layers. The results are four distinct probability distributions, corresponding to the transformations of Hue, Saturation,

Rotation, and Flip. We use them as criteria to compute the loss function, enabling backprop-agation and parameter tuning. Finally, we obtain a set of parameters with high accuracy that can be a perfect fit for the latter training of SWUNet-CD. Therefore, we use this set of parameters as a pre-trained model to replace the normal one, hoping to achieve better experimental results for change detection in the research area.

## 4. Experimental Results

### 4.1. Experimental Settings

The initial learning rate for model training is set to $3 \times 10^{-4}$ The exponential decay learning rate schedule is used, and the learning rate is reduced to 95% every ten epochs. The total number of epochs is 200. One of the most famous optimization algorithms, Adam, is chosen to optimize the network's ability. The pre-trained model is obtained through the self-supervised learning method mentioned above. As for the loss, we choose the mean of the Cross-Entropy loss and the Dice loss. The reason is that the foreground has taken a much smaller place than the background, and this disproportion will lead to more missing detection problems. Therefore, we apply the Dice loss to solve this problem and extract more foreground information. However, the prediction error of some pixels will cause the Dice loss to fluctuate wildly and lead to bad results, so the Cross-Entropy loss that averages the whole is used to reach a compromise. In addition, this experiment uses a large number of data augmentation methods, which can play an important role in avoiding overfitting, improving the robustness of the model, and improving the expression ability of the model. The following methods are mainly used: random cropping with a crop size of 512 pixels × 512 pixels, random flip, random transposition, random HSV transformations that enhance the three dimensions of the color space (hue, saturation, value), random affine transformations including translation, deflation, and rotation, random optical distortion, and random Gaussian noise. All the data augmentation functions above are applied with a probability of 0.3. In addition, our experiment also adopts the method of training weight decay, which is set to $1 \times 10^{-3}$. The overall experiments are conducted on two RTX 3090 GPUs (24 GB memory).

### 4.2. Accuracy Evaluation

The prediction results on the test dataset usually produce four different types of pixels, namely, the truly positive pixels (TP), the false negative pixels (FN), the truly negative pixels (TN), and the false positive pixels (FP), as shown in Table 2. To evaluate the results, we adopt four widely recognized accuracy evaluation metrics, which are calculated by different strategies using the above four sorts of pixels. Precision Rate (Pre) represents the ratio of the predicted real-change pixels to the predicted overall change pixels (Formula (6)). The Recall Rate (Rec) represents the ratio of the predicted real-change pixels to the actual overall change pixels (Formula (7)). The F1 score is the harmonic mean of the contradictory precision evaluation indicators of Pre and Rec, which is used to combine both to make an overall evaluation (Formula (8)). The Intersection over Union (IoU) represents the ratio of the intersection to the union of the real-change pixels and the predicted change pixels, which measures the similarity between the predicted situation and the actual situation (Formula (9)). Each index may focus on different performances through various methods, so they need to be judged comprehensively.

$$Precision = \frac{TP}{TP + FP}, \tag{6}$$

$$Recall = \frac{TP}{TP + FN}, \tag{7}$$

$$F_1 = \frac{2 \times Precision \times Recall}{Precision + Recall} = \frac{2TP}{FP + FN + 2TP}, \tag{8}$$

$$IoU = \frac{Precison \times Recall}{Precision + Recall - Precision \times Recall} = \frac{TP}{FP + FN + TP}, \tag{9}$$

**Table 2.** four kinds of pixels.

|  | Label True | Label False |
|---|---|---|
| **Test True** | TP (True Positive) | FP (False Positive) |
| **Test False** | FN (False Negative) | TN (True Negative) |

*4.3. Ablation Study*

In this section, we devise an ablation study to explore and verify the influence of different factors on the performance of the proposed network. The results of the ablation experiments are shown in Table 3, where the bold values represent the maximum values in the same index. We also predict six groups of images to help show our network's performance, each of which represents a representative set of test data. Four colors are used to make the different pixels clearer: (1) the white part represents the TP pixel; (2) the black part represents the TN pixel; (3) the red part represents the FN pixel; (4) the blue part represents the FP pixel.

The overall ablation study includes the following two aspects: one is to explore the function of the improved pooling module SPPM compared with the traditional pooling layer, the traditional multi-scale fusion modules such as PPM and ASPP through experiments, as shown in Figure 5. The other one is to verify the performance of the pre-trained model obtained through the self-supervised network in comparison with the normal pre-trained model, as shown in Figure 6.

**Table 3.** Ablation Study.

| Backbone | PPM | ASPP | SPPM | Self-Supervised | Pre | Rec | F1 | IoU |
|---|---|---|---|---|---|---|---|---|
| ✔ |  |  |  |  | 0.6676 | 0.6834 | 0.6754 | 0.5099 |
| ✔ | ✔ |  |  |  | 0.6722 | 0.6975 | 0.6846 | 0.5205 |
| ✔ |  | ✔ |  |  | 0.6856 | 0.6746 | 0.6801 | 0.5152 |
| ✔ |  |  | ✔ |  | 07223 | l0.7136 | 0.7179 | 0.5600 |
| ✔ |  |  |  | ✔ | 0.7116 | 0.6777 | 0.6942 | 0.5317 |
| ✔ |  |  | ✔ | ✔ | **0.7340** | 0.7133 | **0.7235** | **0.5668** |

4.3.1. Effects of Different Pooling Modules

This section mainly compares the accuracy of four different experimental results obtained with no pooling layer, with PPM, with ASPP, and with SPPM. No pooling layer is set as the base case, it can be used to more intuitively observe the ability of different pooling modules to improve the network's performance. PPM, ASPP, and the proposed SPPM are all advanced pooling modules that employ the idea of multi-scale fusion. Their difference mainly lies in the method of sampling. PPM is first proposed in the PSPNet for semantic segmentation tasks, which acquires multi-scale features through four adaptive average pooling layers of different sizes. As for ASPP, it is firstly proposed in the DeepLabV2 for semantic segmentation tasks, which extracts multi-scale features by constructing four atrous convolutional layers of unequal sizes. Much different from the previous methods, we make a difference by using a transformer. Our SPPM builds four W-MSA/SW-MSA feature extraction layers of different sizes to obtain multi-scale features. Considering the great performance of the transformer, we think this improvement will add to the ability of our network. Therefore, we conduct ablation experiments using the GF-1 dataset in Hebei Province for the above four cases to help prove its performance and verify our thoughts.

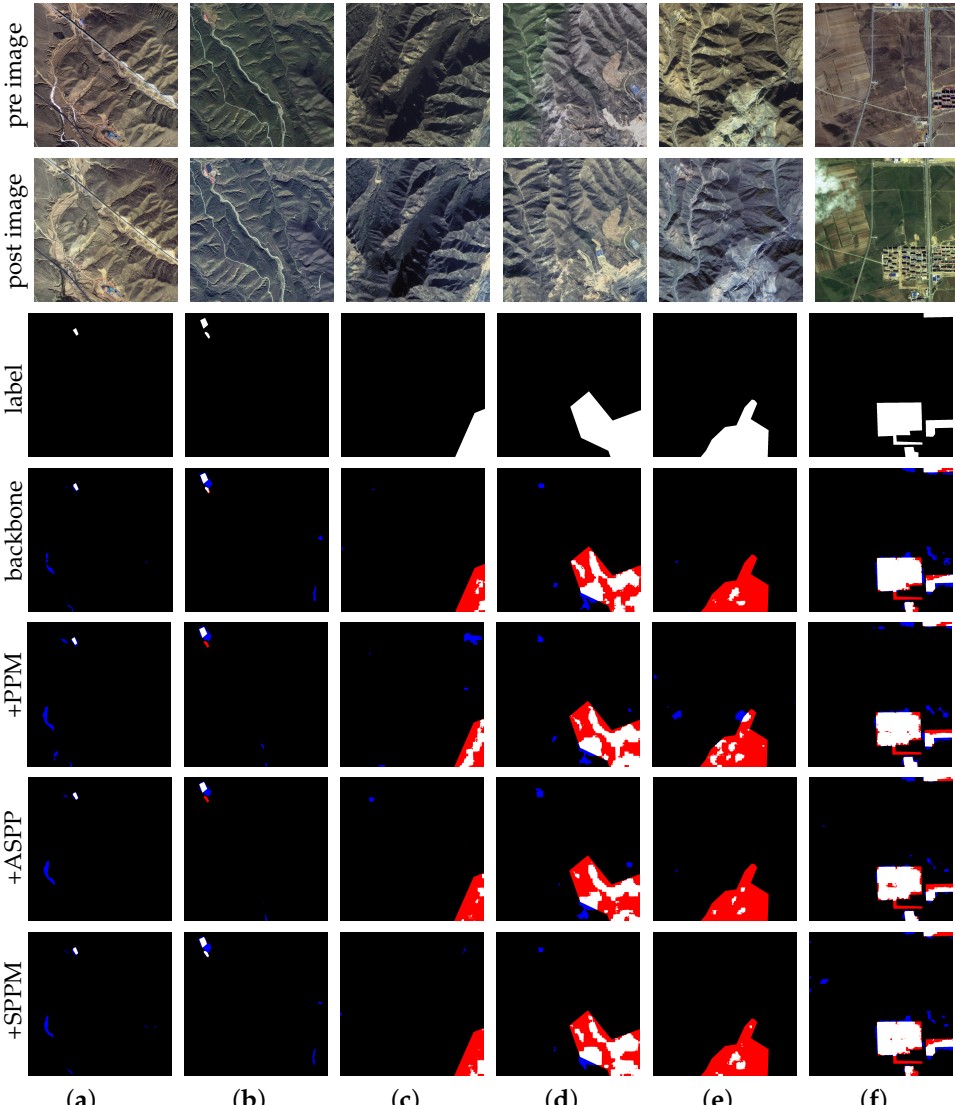

**Figure 5.** Six groups of images using different pooling methods. (**a**) Buildings in Eco-redline areas. (**b**) Buildings in Eco-redline areas. (**c**) Mines in Eco-redline areas. (**d**) Mines in Eco-redline areas. (**e**) Mines in Eco-redline areas. (**f**) Buildings in rural areas.

It can be seen from Table 3 that all the networks using the multi-scale fusion pooling module have significantly improved their accuracy compared to the case without the pooling module. Among them, PPM and ASPP have mutual advantages and disadvantages in the improvement of precision rate and recall rate. PPM has a higher recall rate, while ASPP has a higher precision rate, but both of them have improved the whole accuracy. However, through the bold numbers, it is not difficult to see that SPPM is much better at improving the network in both precision rate and recall rate, both surpassing 0.7, while the previous situation is all below 0.7. Correspondingly, its F1 score and IoU are also higher, reaching 0.7179 and 0.5600. Based on these evaluation indicators, we can partly conclude that the pooling module can help improve the network's ability in our change detection task, and our SPPM has a more obvious improvement in the network's overall performance than PPM and ASPP.

Moreover, we can also see intuitively from Figure 5 that the network with PPM has fewer red parts and more blue parts, which represents its relatively high recall rate and relatively low precision rate. While with ASPP, the results have more red parts and fewer blue parts, which represents its relatively low recall rate and relatively high precision rate.

As a matter of fact, our SPPM has both fewer red parts and blue parts, which also proves its good performance.

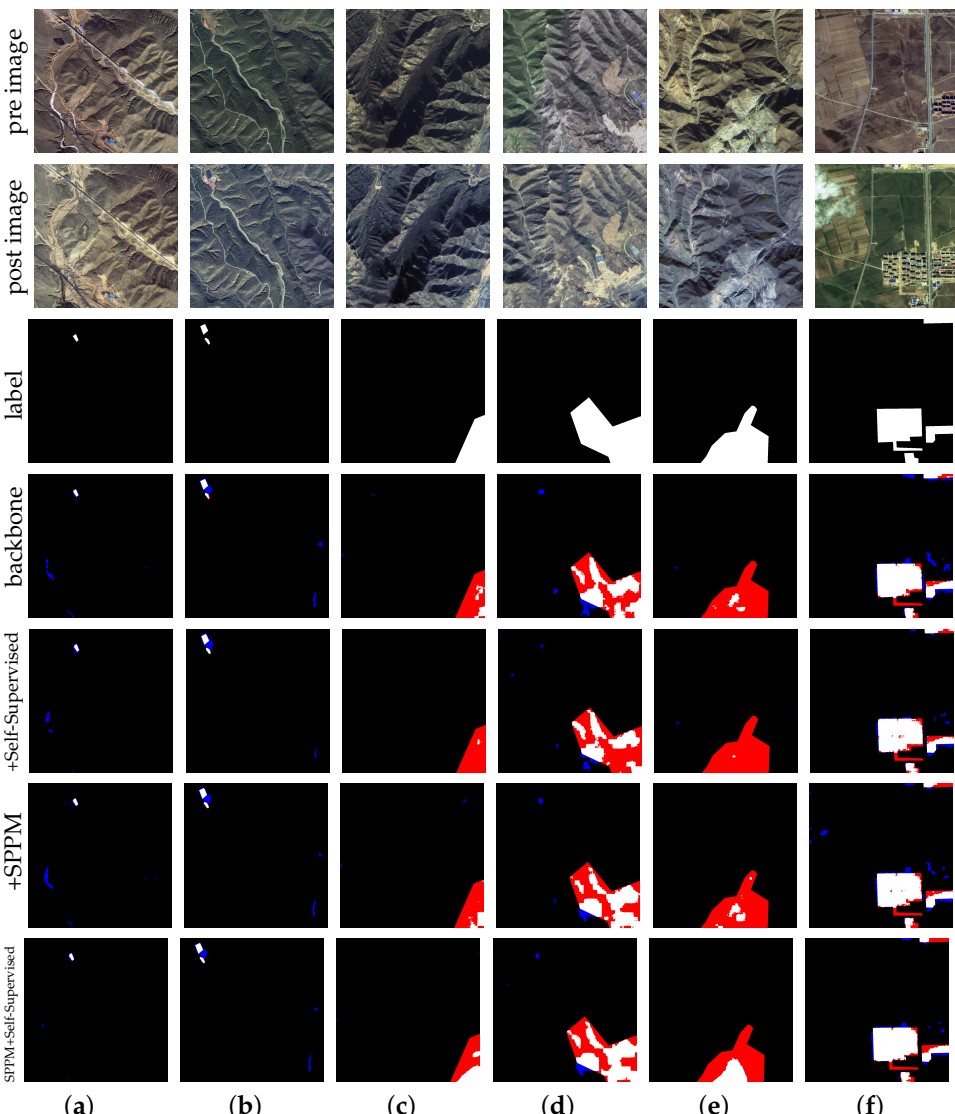

**Figure 6.** Six groups of images using different pre-trained models. (**a**) Buildings in Eco-redline areas. (**b**) Buildings in Eco-redline areas. (**c**) Mines in Eco-redline areas. (**d**) Mines in Eco-redline areas. (**e**) Mines in Eco-redline areas. (**f**) Buildings in rural areas.

### 4.3.2. Effects of Self-Supervised Study

This section mainly compares and analyzes the ability of our network using different pre-trained models, the normal one and the one obtained by the self-supervised learning method mentioned above. The overall experiments are carried out in two distinct environments: the basic network and the network with SPPM. As a matter of fact, the differences between the two pre-trained models lie mainly in two aspects: one is about the data source, and the other one is about the supervised method. As for data source, the normal pre-trained model is acquired by training public datasets, while our pre-trained model is obtained by a self-supervised network using GF-1 remote sensing images in Hebei Province. It is obvious that our dataset directly focuses on the research areas, which can help our network function better on our research target. However, from the perspective of supervised methods, the former pre-trained model has an advantage over the latter. Since the public dataset is of high quality and has good labels, it adopts the supervised learning method.

Our pre-trained model is obtained using the unsupervised method, with remote sensing images without labels. Therefore, both models have mutual advantages and disadvantages, and is hard to tell which performs better.

So we calculate the accuracy and do the prediction, according to Figure 6 and Table 3. The result turns out that whether it is the backbone or the network using SPPM, the accuracy values after using our pre-trained model are significantly improved compared with the values acquired using the normal pre-trained model. SWUNet-CD with SPPM and Self-supervised method even reaches 0.7340 in precision rate and 0.7133 in recall rate, the best result and the second best result. This result also helps to explain that our self-supervised method helps the network identify more changes in the research area, and its lack of label also leads to the problem of more wrong predictions. In fact, since its F1 score and IoU both reach the highest, it demonstrates that our self-supervised learning method can further optimize our experimental task. Moreover, the less blue and red parts compared with others in Figure 6 also help to prove the demonstration.

In general, the ablation study in this section conducts experiments and analytical verifications on the role of the pooling layer and the impact of the self-supervised learning method and finally draws the following conclusions:

- SPPM combining W-MSA/SW-MSA and the idea of multi-scale fusion can not only effectively improve the performance of the basic network but also has significant advantages compared to other multi-scale fusion modules such as ASPP and PPM.
- Self-supervised learning method can obtain a pre-trained model that is more suitable for the research area than the normal pre-trained model, which can further improve the performance of the network and optimize the prediction results without adding additional data labeling work.

### 4.4. Comparison with Other Neural Networks

To better verify the effect of SWUNet-CD on the target change detection task, some related well-performed change detection networks are chosen for comparison. These networks are as follows:

- Fully Convolutional Early Fusion (FC-EF) [43,44]: The most basic data-level fusion change detection network. It is constructed based on the U-Net structure, and the data of different phases are fused before entering the network.
- Fully Convolutional Siamese Concatenation network(FC-Siam-Conc) [43,44]: A Siamese change detection network designed based on FC-EF. This network uses the idea of feature-level fusion and builds on the structure of U-Net. It concatenates different feature maps obtained after down-sampling in the channel dimension.
- Fully Convolutional Siamese Difference network(FC-Siam-Diff) [43,44]: A Siamese change detection network designed based on FC-EF. Different from FC-Siam-Conc, it pays more attention to the difference of the images and does not stitch the feature maps but calculates their absolute difference.
- Deeply-Supervised image fusion network (DSIFN) [45]: A change detection network for high-resolution remote sensing images. It uses VGG-16 as the backbone to down-sample and obtain the image's features and then fuses multi-level depth features and image features through a difference discrimination network using an attention mechanism.
- CDNet [46]: A network first used in building change detection. It combines point cloud data of different phases with orthophotos. The two inputs of the final network are the height differences of ALS-DSM and DIM-DSM, and the corresponding orthophoto data.
- BiT [47]: A small and efficient change detection network. It uses the improved ResNet-18 as the backbone [48], designs, and adds a bitemporal image Transformer between the upsampling and downsampling modules. Its aim is to extract the real difference between high-dimensional semantic tokens.

- Siamese NestedUNet (SNUNet-CD) [49]: A Siamese network for change detection designed and proposed based on the idea of NestedUNet [50]. It has two main characteristics, which are listed as follows: (1) there exist highly dense connections distributed in the network, including a large number of skip connections between encoders and decoders. Such a structure can allow it to ensure high-resolution and fine-grained representation, and alleviate the loss of localization information in the deep layers of the neural network. (2) A deep monitoring method with an ensembled channel attention module (ECAM) is proposed. This module can aggregate and refine features from multiple semantic levels, suppressing semantic gaps and localization errors to a certain extent.

We conduct various experiments and compare the results of these seven outstanding change detection networks with our network, SWUNet-CD, which aims to better verify the performance of our method on the target task. Moreover, all networks are performed under the same experimental conditions and hyperparameters to obtain the best comparative results.

Table 4 shows the accuracy evaluation results obtained by different networks, and Figure 7 exhibits six groups of selected representative change detection results in the prediction dataset.

**Table 4.** Comparison with other networks.

| Network | Precision | Recall | F1 Score | IoU |
|---------|-----------|--------|----------|-----|
| FC-EF | 0.6621 | 0.6596 | 0.6608 | 0.4935 |
| FC-Siam-Conc | 0.6857 | 0.6633 | 0.6743 | 0.5087 |
| FC-Siam-Diff | 0.7072 | 0.6678 | 0.6869 | 0.5232 |
| DSIFN | 0.8140 | 0.6284 | 0.7093 | 0.5495 |
| CDNet | 0.7081 | 0.6913 | 0.6996 | 0.5380 |
| BiT | 0.7237 | 0.6940 | 0.7085 | 0.5486 |
| SNUNet-CD | **0.7344** | 0.6719 | 0.7018 | 0.5405 |
| SWUNet-CD | 0.7340 | **0.7133** | **0.7235** | **0.5668** |

Table 4 summarizes the four accuracy evaluation indicators of all experiments. The maximum value of each accuracy evaluation index is marked in bold. It can be seen intuitively from this that since the early three FC network structures are relatively simple, their accuracy is generally lower than that of the later networks. However, the accuracy of FC-Siam-Conc and FC-Siam-Diff, which use feature-level fusion, are also significantly improved compared to FC-EF, which uses data-level fusion. The other five networks with more complex structures all have more significant improvements in various indicators. Overall, their precision rate, recall rate, F1 score, and IoU all generally reach 0.70, 0.67, 0.70, and 0.54, respectively. From a specific analysis, it is not difficult to see that although BiT has the highest accuracy in the precision rate, reaching 0.8140, the highest accuracy value of the other three indexes is all achieved by the SWUNet-CD. In fact, BiT does achieve a higher precision rate, but its recall rate only reaches 0.6284, which is the lowest among all networks. It means that BiT's missing detection problem is very serious, and its F1 score and IoU also intuitively reflect the shortcomings of the BiT. Moreover, other complex networks are also inferior to SWUNet-CD in various accuracy indicators. To be specific, SWUNet-CD achieves 0.7 evenly in both precision rate and recall rate, and also has significant advantages over other networks in F1 score and IoU, which intuitively reflects its superior performance in our experiments.

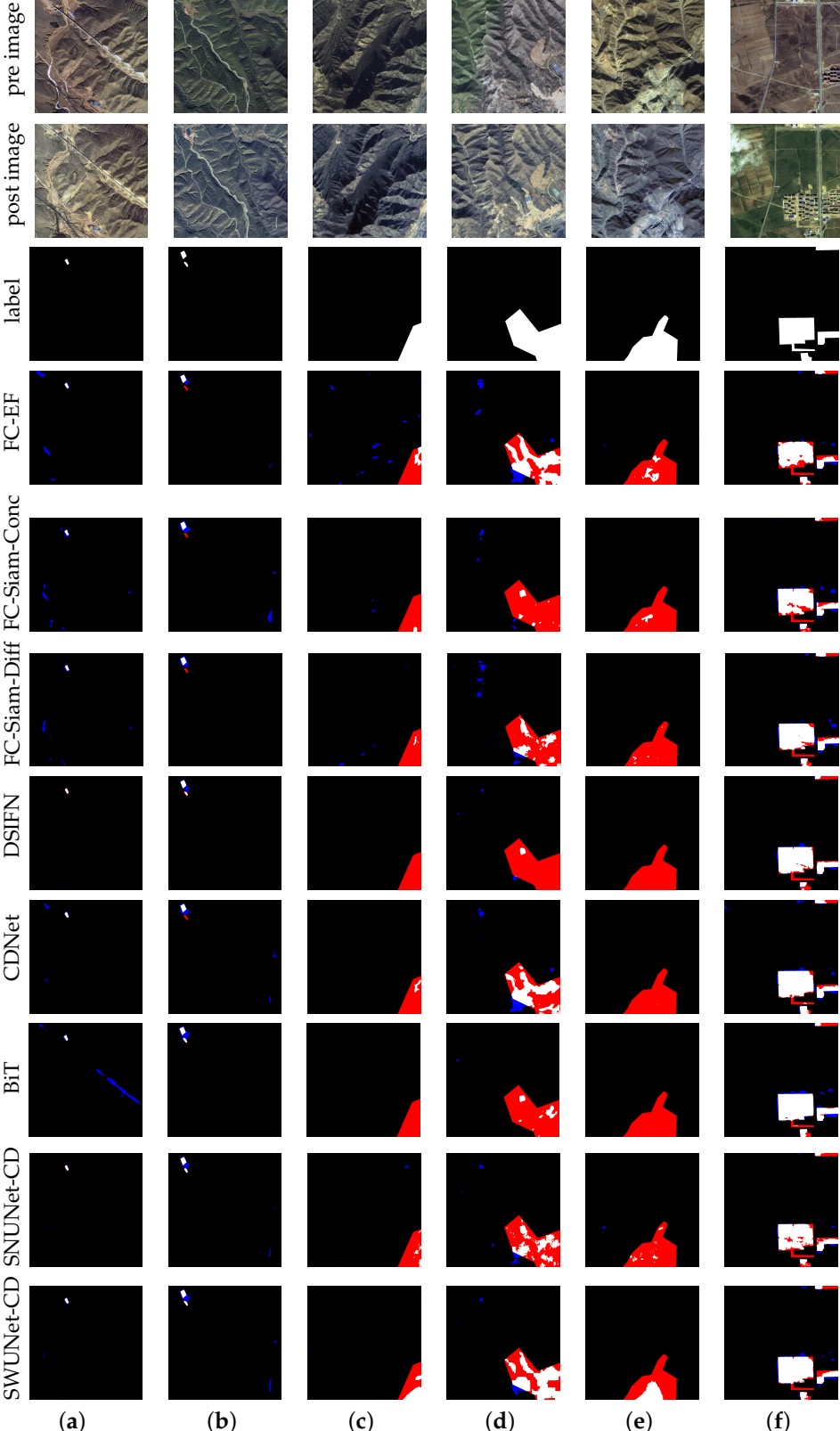

**Figure 7.** Six groups of images using different change detection networks. (**a**) Buildings in Eco-redline areas. (**b**) Buildings in Eco-redline areas. (**c**) Mines in Eco-redline areas. (**d**) Mines in Eco-redline areas. (**e**) Mines in Eco-redline areas. (**f**) Buildings in rural areas.

Figure 7 has six columns in total, and each represents a representative set of test data. The first two rows represent pre-phase data and post-phase data, respectively. The third row represents the corresponding change patch. Starting from the fourth row are the prediction results of each network. To show the results more intuitively, we use four colors to represent different predictions of pixels. The white part represents the TP pixel, the black part represents the TN pixel, the red part represents the FN pixel, and the blue part represents the FP pixel. Each group of data has been arranged according to different objects of interest. The first two columns mainly focus on the changes in illegal buildings in the ecological protection red line area: the demolition of illegal buildings appears in the first set of data. It can be seen that all networks are basically able to detect changes well, but our network has almost no false detections. The second set of data mainly focuses on the construction of illegal buildings. Some networks are unable to accurately identify the new buildings, while the other networks that are able to detect the changes have a more serious problem of false detection. Our network seems to perform best among them. The following three columns mainly focus on the changes in mining areas and quarries in the Eco-redline areas, which are very hard to detect. The overall identification of the third group is generally poor, and the problem of missed detection is very serious. Some networks nearly fail to detect the changes. Although our method also misses a large number of changes, it still provides the best prediction result among all networks by comparison. In the fourth group, from the size of the blue parts and red parts, it is not difficult to judge that our network has better performance considering both missed detections and false detections. Then in the fifth group, our network can still identify a part of the whole mining area while the other networks are almost unable to do the job. At the same time, there exists nearly no false detection problem, which proves that our network shows a significant advantage in this set of cases. The sixth column is about the building change in a cultivated area. This set of data is to test whether the model obtained in our experiment can also have a great ability to identify changes within the non-Eco-redline areas. From this column, it can be seen intuitively that SWUNet-CD also has a better recognition effect for changes in other regions.

Combining the results of Table 4, Figure 7, and the analysis above, we can firmly draw the following conclusions: (1) SWUNet-CD can detect more changes than other networks, especially the changes in mining areas which are very hard to detect. (2) SWUNet-CD can also ensure not to make more false detections. To be specific, it means that while ensuring a higher precision rate, the recall rate is also significantly improved so that the F1 score and IoU can be much higher when compared to other networks.

*4.5. Intersection Analysis of Network Extracted Spots and Manually Extracted Spots*

Table 5 and Figure 8 can be obtained by predicting some large-scale remote sensing images in the Eco-redline areas and making an intersection analysis between the results and the corresponding manually extracted objects. This evaluation result reflects the network's ability to locate the changing area, which verifies the validity of this experiment from another perspective. Therefore, we choose our network and three other well-performed networks to test and make a comparison, and the results are presented in a table and four figures. Table 5 records the overall intersection of prediction results and labels, in which ✔ represents the intersection area while ✘ represents the non-intersect area. Figure 8a shows the overall extent of the test region. The following four figures produce a more intuitive result, in which the green spots represent the detected change areas while the red spots represent the undetected change areas. The detailed information is as follows: Figure 8b represents the results of SWUNet-CD. There are a total of $411 + 39 = 450$ manually extracted spots, 411 correct spots extracted by the network, and the recall rate is 91.33%. Figure 8c represents the results of SNUNet-CD. There are a total of $389 + 61 = 450$ manually extracted spots, 389 correct spots extracted by the network, and the recall rate is 86.44%. Figure 8d represents the results of DSIFN. There are a total of $388 + 62 = 450$ manually extracted spots, 388 correct spots extracted by the network, and the recall rate is

86.22%. Figure 8e represents the results of BiT. There are a total of $391 + 59 = 450$ manually extracted spots, 391 correct spots extracted by the network, and the recall rate is 86.89%. According to the data, SWUNet-CD has also surpassed other networks in this study, which strongly supports its performance in practical application.

**Table 5.** Intersection Recall Calculation.

| Network | Prediction | Label | Spots Number | Proportion | Recall |
|---|---|---|---|---|---|
| SWUNet-CD | ✔ | ✔ | 411 | 91.33% | 91.33% |
| | ✘ | | 23 | | |
| | | ✘ | 39 | 8.67% | |
| SNUNet-CD | ✔ | ✔ | 389 | 86.44% | 86.44% |
| | ✘ | | 26 | | |
| | | ✘ | 61 | 13.56% | |
| DSIFN | ✔ | ✔ | 388 | 86.22% | 86.22% |
| | ✘ | | 24 | | |
| | | ✘ | 62 | 13.78% | |
| BiT | ✔ | ✔ | 391 | 86.89% | 86.89% |
| | ✘ | | 28 | | |
| | | ✘ | 59 | 13.11% | |

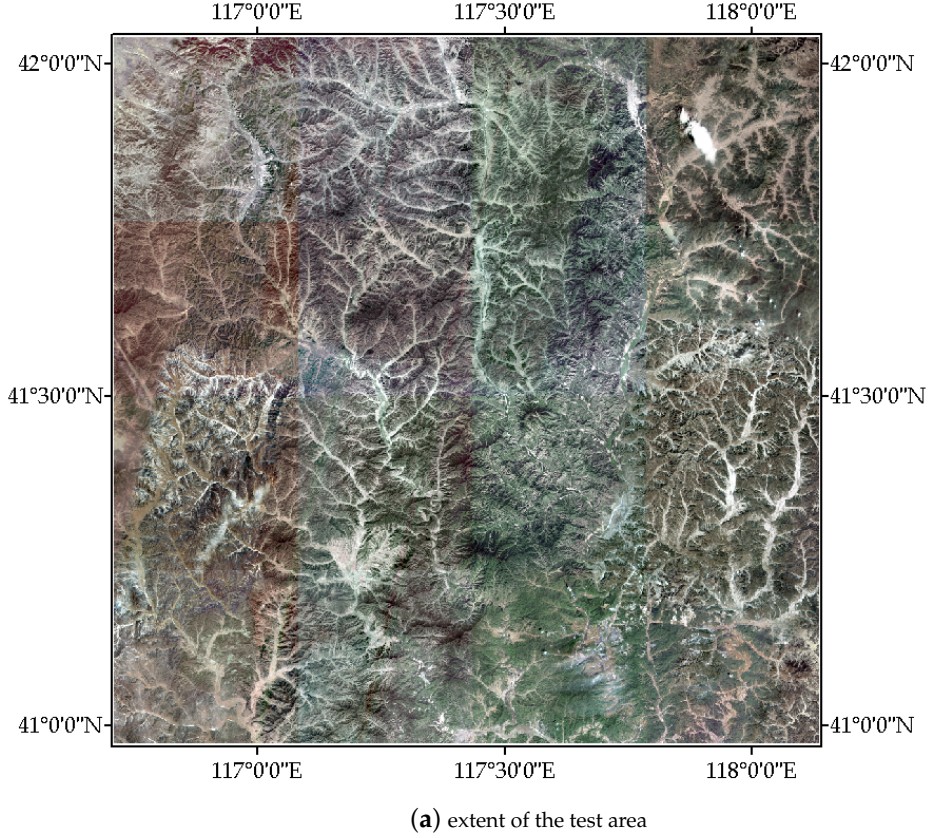

(**a**) extent of the test area

**Figure 8.** *Cont.*

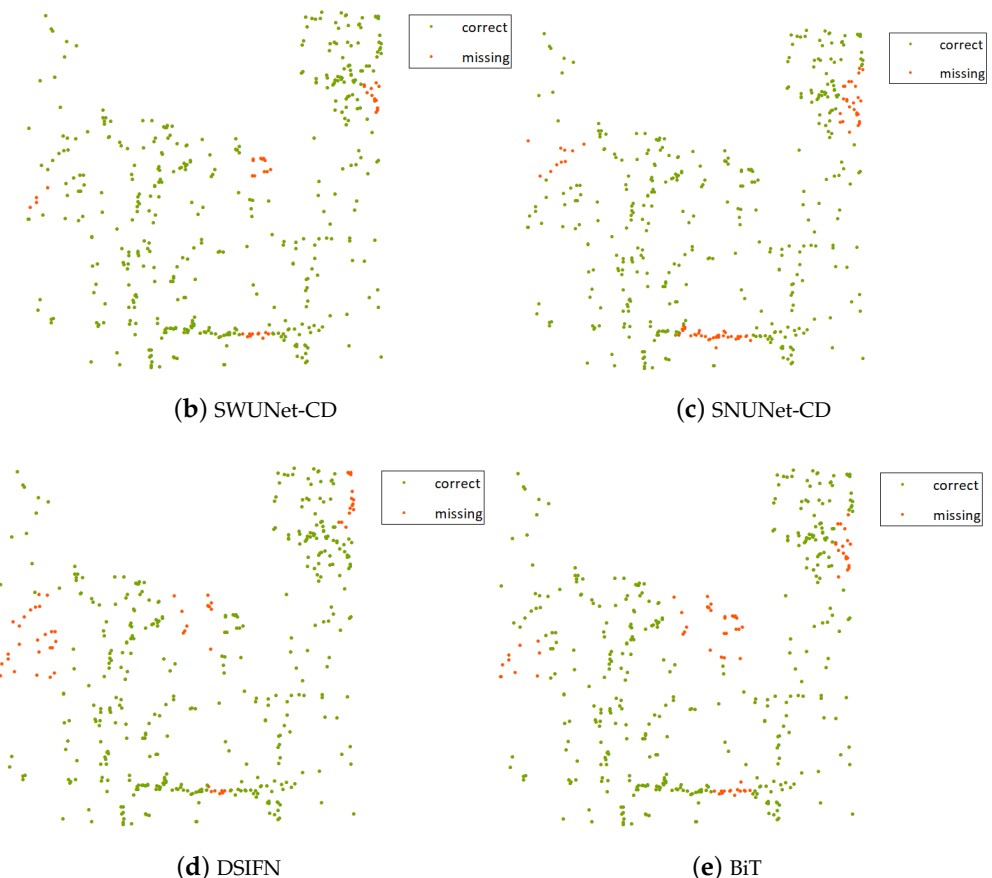

**Figure 8.** (**a**) The extent of the test area. (**b**) Intersection spots by SWUNet-CD. (**c**) Intersection spots by SNUNet-CD. (**d**) Intersection spots by DSIFN. (**e**) Intersection spots by BiT.

## 5. Conclusions

The change detection method combining remote sensing data with neural networks is a useful and necessary research direction in the current society. At present, it has been widely used in urban construction, disaster assessment, nature protection, and other fields. In our study, we used this kind of method to detect changes in Eco-redline area, aiming to meet the urgent needs of environmental monitoring.

To get better results, we have adopted three methods. Firstly, we constructed a Siamese network called SWUNet-CD. This structure will pay more attention to the use of deeper features in remote sensing images, which will help to learn the differences that cannot be easily identified. Secondly, we devised a special pooling module called SPPM. This module is used to maintain features of different levels, which will ensure that the network does not miss any key information. Finally, we constructed a self-supervised network based on the contrastive method, and trained it to obtain a pre-trained model. This model is used to partly solve the uneven distribution problem due to large negative samples and small positive samples, and will be more suited to remote sensing research. It can preliminarily be seen from the experimental results that our research method has achieved good results, reaching an F1 score of 0.7235 and 0.5668 in IoU, which is much better than the baseline and other comparable networks. Moreover, we continue to verify the performance through practical ways, predicting and calculating intersection spots which can serve as the direct basis of Eco-redline monitoring. By reaching 91.33% in its intersection recall rate, this stage also proves the great performance of our method.

However, there still exist some drawbacks that cannot be overlooked: (1) the change detection results of mining areas are much worse than the results of the buildings. To solve the problem, we have thought about two ways. Firstly, we can investigate more about the

types of mining areas. For example, some are located on the ground while others are below, and some may be accompanied by water. So it may be very meaningful to make a precise subdivision about the mining areas. Secondly, we can improve the self-supervised network. By manually adding some mining areas to the image, we can insert some supervised information, helping to improve the network's ability to identify the mining areas. (2) The performance of the pre-trained model. Although we have thought of ways to obtain a model which better fits the remote sensing images and our change detection task, it still lacks a degree of change information, and, therefore, its ability may be prohibited. (3) The quality and the usability of the data. We need more supervised data to train the network, but the labeling work is a huge and hard task and will be easily disturbed by the nearby environment.

Therefore, our future research will mainly focus on the following aspects: (1) doing more research about the type of mining areas and applying it in the labeling work; (2) studying other self-supervised methods and trying to obtain a pre-trained model, especially, for change detection task; (3) bringing in other datasets and testing our network's ability to detect changes in other research areas like rural areas and urban areas.

**Author Contributions:** Conceptualization, Y.Z.; methodology, Y.Z.; validation, X.Y. and P.C.; formal analysis, L.Z.; resources, T.S.; data curation, P.C. and X.Y.; writing—original draft preparation, P.C. and Y.Z.; writing—review and editing, Y.Z.; visualization, Y.Z.; supervision, P.C.; project administration, Z.C.; funding acquisition, Z.C. All authors have read and agreed to the published version of the manuscript.

**Funding:** This research was funded by the National Key Research and Development Program of China (Grant No. 2021YFB390110302).

**Institutional Review Board Statement:** Not applicable.

**Informed Consent Statement:** Not applicable.

**Data Availability Statement:** Not publicly available.

**Acknowledgments:** The authors are grateful to the editors and anonymous reviewers for their informative suggestions.

**Conflicts of Interest:** The authors declare no conflict of interest.

## Abbreviations

The following abbreviations are used in this manuscript:

| | |
|---|---|
| ECL | Ecological redline |
| CNN | convolutional neural network |
| ViT | Visual Transformer |
| Swin | Shifted Window-based Self-attention |
| FN | False negative |
| FP | False positive |
| TP | True positive |
| TN | True negative |
| PPM | Pyramid Pooling Module |
| ASPP | Atrous Spatial Pyramid Pooling |
| IoU | Intersection over Union |
| SWUNet-CD | Siamese Swin-UNet for change detection |
| FC-EF | Fully Convolutional Early Fusion |
| FC-Siame-Conc | Fully Convolutional Siamese-Concatenation |
| FC-Siam-Diff | Fully Convolutional Siamese-Difference |
| Swin-T | Hierarchical Vision Transformer using Shifted Windows |
| Swin-UNet | Unet-like Pure Transformer for Medical Image Segmentation |
| DSIFN | Deeply supervised image fusion network for change detection |
| SNUNet-CD | Siamese NestedUNet |
| AIFM | Attention Information Module |

| ECAM | Ensembled Channel Attention Module |
| PCA | Principal component analysis |
| ICA | Independent component analysis |
| MAD | Multivariate alteration detection |
| SPPM | Swin-based Pyramid Pooling Module |

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
