# Peer review of "A Transformer-Based Neural Network with Improved Pyramid Pooling Module for Change Detection in Ecological Redline Monitoring"

_remotesensing, doi:10.3390/rs15030588_

Round 1
Reviewer 1 Report
General comments
The authors propose a transformer-based neural network method with improved pyramid pooling module for change detection in ecological redline monitoring. Remote sensing algorithms for change detection of land cover are already abundant, but the accuracy reported in this study was not significantly high and computational efficiency was not quantified. I suggest that authors re-think what exact contributions of this study on the change detection of underlying surface.
Specific comments
1. This study describes the significance of ecological redline in China, while the method reported by this study is specifically designed to detect changes of land cover. The authors should carefully consider the whole aims and applicability of the study. If the method can be used in other regions of outside of ecological redline regions?
2. The relevant literatures lack for supporting the shortcomings and selection of data-level fusion, feature-level fusion, and decision-level fusion mentioned in Line 77-89.
3. The Recall, Precision, and F-score in Table 4 are all low and insufficient to indicate the accuracy of this method. In addition, the accuracy suddenly improves in Table 5, why?
4. The sample labels of change within the study area were selected in a more concentrated manner and only one or several samples were available in each area. The extent of detection of change will affect the final accuracy valuation.
5. In section 4.5, the study conducts experiments in a large-scale region. Can you add some image and geographic information of the region?
6. The self-supervised training model proposed in the study is not significantly different from the results obtained by the conventional model. So what kind of improvements should be made? It would be nice to add further improvement ideas.
Reviewer 2 Report
This research provides a new approach to ecological remote sensing based on a deep learning model to solve the problem of ecological red line monitoring application, which has good application value. However, there are several issues that need further improvement.
(1) The self-supervised learning method need to be described more specifically, including your reason and criterion for HSV transformation, rotation and flip. Also, I would like to know why you choose hue and saturation as pseudo label, but put value aside.
(2) The description of each module in section 3, the description of data augmentation in section 4, and the description of other networks in section 4 need to be written in a more organized way, instead of just putting all words in one paragraph.
(3) You could consider adding more analysis of your comparative experiment in section 4.
(4) The abstract needs to be further fleshed out to provide more critical information. The method proposed in the manuscript is superior, but no specific data are provided on how much the accuracy is improved and to what extent the performance is improved.
Reviewer 3 Report
This manuscript examined the performance and effectiveness of a transformer-based neural network in detecting changes in ecological redline over Hebei Province. Overall, this manuscript is well-written, and the experimental design is reasonable. Detailed analysis of their results are also provided. I don’t have any major concern, but a few minor issues:
Line 169-170: What do you mean by “different phases”? You should provide the detailed time information (e.g., which year for each phase).
Line 177: change “nice” to “typical” or “good”
Line 195-197: How did you convert the four-band image into a three-band image? Which method do you use? You should make your description clearer.
Round 2
Reviewer 1 Report
The authors has revised all my concerns. It would be better if the accuracy can be further improved.
Round 3
Reviewer 1 Report
Considering that the authors have answered my previous concerns, it is recommended to accept.